# Liquid Biopsy in Cancer: Focus on Lymphoproliferative Disorders

**DOI:** 10.3390/cancers14215378

**Published:** 2022-10-31

**Authors:** Francesco D. Savino, Fabio Rigali, Viviana Giustini, Deborah D’Aliberti, Silvia Spinelli, Rocco Piazza, Antonio Sacco, Aldo M. Roccaro

**Affiliations:** 1Clinical Research Development and Phase I Unit, ASST Spedali Civili di Brescia, 25121 Brescia, Italy; 2Clinical Chemistry Laboratory, Flow Cytometry Unit, ASST Spedali Civili di Brescia, 25121 Brescia, Italy; 3Department of Medicine and Surgery, University of Milano-Bicocca, 20900 Monza, Italy; 4Hematology and Clinical Research Unit, San Gerardo Hospital, 20900 Monza, Italy

**Keywords:** liquid biopsy, lymphoproliferative diseases, onco-hematology

## Abstract

**Simple Summary:**

Liquid biopsy (LBx) is a novel and promising approach in precision medicine, suitable for patient management in a wide range of medical conditions. Its utility in oncology ranges from disease screening to early diagnosis and treatment. LBx has several strengths, such as safeness, quickness of execution, and repeatability, compared to old-fashioned solid biopsy techniques; indeed, LBx requires the collection of a small number of biospecimens. LBx has been proven to be accurate and reliable, as demonstrated in several clinical studies, and it could have a surprising impact on survival and quality of life for cancer patients in the near future. With the present review article, we aim to summarize LBx characteristics, considering both the clinical and the laboratory settings; and to collect the most recent evidence within the oncology field, with a specific focus on blood cancers.

**Abstract:**

Within the context of precision medicine, the scientific community is giving particular attention to early diagnosis and intervention, guided by non-invasive methodologies. Liquid biopsy (LBx) is a recent laboratory approach consisting of a non-invasive blood draw, which allows the detection of information about potential prognostic factors, or markers to be used for diagnostic purposes; it might also allow the clinician to establish a treatment regimen and predict a patient’s response. Since the discovery of circulating tumor cells (CTCs) in the nineteenth century, the possibility of integrating LBx into clinical practice has been explored, primarily because of its safeness and easy execution: indeed, compared to solid biopsy, sampling-related risks are less of a concern, and the quickness and repeatability of the process could help confirm a prompt diagnosis or to further corroborate the existence of a metastatic spreading of the disease. LBx’s usefulness has been consolidated in a narrow range of oncological settings, first of all, non-small cell lung carcinoma (NSCLC), and it is now gradually being assessed also in lymphoproliferative diseases, such as acute lymphocytic leukemia (ALL), B-cell lymphomas, and multiple myeloma. The present review aims to summarize LBx’s overall characteristics (such as its advantages and flaws, collection and analysis methodologies, indications, and targets of the test), and to highlight the applications of this technique within the specific field of B-cell malignancies. The perspectives on how such a simple and convenient technique could improve hemato-oncological clinical practice are broadly encouraging, yet far from a complete integration in routine clinical settings.

## 1. Introduction

Liquid Biopsy (LBx) represents an approach that allows the identification of disease-related biomarkers, through a simple and minimally or non-invasive procedure, such as a blood draw or collection of other body fluids, including saliva, urine, cerebrospinal fluid (CSF), and stool. Particular attention has been given to LBx within the oncology field, in relation to both solid tumors and hematologic malignancies. For instance, research studies have shown how LBx could be used to support the clinician in diagnosis, treatment selection, as well as disease monitoring [1,2,3,4,5,6].

By performing and interpreting an LBx we may get useful information related to the detection of circulating tumor DNA (ctDNA), circulating tumor cells (CTCs), circulating RNAs, and extracellular vesicles (EVs) through a peripheral blood draw.

Technologies for processing, isolating, and collecting tumor material, together with analysis techniques and the discovery of new research targets, have rapidly evolved, especially in the last decade. The first PCR-based test assessed on humans goes back to 2016 with the approval of the FDA, followed by the first two NGS-based assays in 2020 [7,8,9]. The implementation of LBx in the clinical management of lung cancer patients has found its rationale in circumstances where low tumor tissue availability could have hindered the execution of a conventional solid biopsy [10,11,12]. LBx has been implemented in cancer screening [13], with CTCs and ctDNA analysis being used for supporting prognosis assessment, defining minimum residual disease (MRD) status, driving treatment selection, and supporting cancer patient follow-up [14,15]. An additional use for LBx refers to early cancer detection, as shown by the relevance of circulating miRNA and cf-DNA in supporting cancer diagnosis [13,16,17,18,19,20,21].

Compared to solid biopsy, LBx offers several advantages (Table 1). First of all, a blood draw is safer, faster, and cheaper than an image-guided invasive surgical procedure, which requires planning, coordination, and execution by dedicated professionals. Moreover, primary tumor sources usually do not involve metastatic foci, something that is, vice versa, detectable within the bloodstream; the traditional approach is also not easily re-accessible through time without exposing the patient to risks due to repeated biopsies. Moreover, procedure-related bleeding and pain may represent common concurrent complications of solid biopsy [16,22,23]. Besides the static information that can be acquired through an LBx (i.e., diagnostic biomarkers), the latter sounds appealing when it comes to monitoring dynamic parameters, such as the activity of the disease and the longitudinal treatment response. Somatic mutations are also variable parameters, given the changing nature of the tumor and metastasis genetic makeup, and are more likely to be investigated through circulating tumor-derived particles [24,25]. LBx could also reveal subclonal mutations responsible for both tumor progression and drug resistance [15].

Finally, recent studies have highlighted the relevance of extracellular vesicles (EVs), for instance, exosomes, which could serve as a crucial source for developing novel application fields for LBx [26]. Circulating EVs present with unique surface markers and disease-specific content, thus leading to the development of biopanels related to specific diseases, functions, and other variables useful in several clinical settings. New technologies are continuously emerging to isolate, identify, and link distinct EV subpopulations to peculiar biological and clinical features, even at an individual and personalized level, as in the case of the innovative phage display approach recently described by Maisano et al. [26]. Once LBx becomes well integrated into the clinical practice, it could replace solid biopsy for those cases requiring multiple draws at close range over time, or for patients that are unfit for undergoing an invasive solid biopsy procedure. Patients unfit for a solid biopsy are, for example, immunosuppressed subjects, who may suffer more easily from procedure-related complications, thus contributing to lower diagnostic power and higher mortality rates [27,28].

The main issue with the application of LBx in clinical practice is the confidence level of the tests and pre-analytic phases. For instance, LBx would not represent the best approach to identifying genetic alterations at an early disease stage, because of the low ctDNA concentration, resulting from a low tumor burden. The same concerns would apply within the MRD setting [20,29,30].

In the present article, we consider LBx from the laboratory and the clinical point of view, focusing on the most recent discoveries about its application within the specific field of lymphoproliferative diseases.

## 2. Liquid Biopsy: New Techniques and Biomarkers

Performing an LBx allows physicians and scientists to investigate tumor cells at both the cellular and molecular level: for instance, ctDNA and CTCs are the most studied so far. Recent studies have also included miRNAs, methylation markers, and extracellular vesicles, as emerging and promising alternatives, widening the diagnostic and therapeutic possibilities of LBx (Figure 1).

### 2.1. cfDNA, ctDNA, and Circulating RNA

Cell-free DNA (cfDNA) consists of total extracellular DNA detectable within the bloodstream, as well as in other biospecimens, such as cerebrospinal and serous fluids [31,32,33].

Cancer cells, as well as non-tumor cells, could shed their genetic material into the bloodstream, as circulating tumor DNA (ctDNA) [34], which has been proven to be useful in supporting diagnosis, prognosis, disease progression, MRD evaluation, and treatment response, within the context of non-small cell lung cancer (NSCLC) and other solid tumors [10,11,12,13,14,15,29,34,35].

cfDNA is mainly derived from hematopoietic cells circulating in the bloodstream [36], where its amount is typically limited and poor in terms of quality. It is often necessary to proceed with amplification processes, which may cause artifacts at both PCR and sequencing levels. Only a small fraction of cfDNA is represented by ctDNA, posing a challenge in terms of isolation, processing DNA, and analysis of the generated data. Moreover, discriminating relevant DNA variants from leucocyte hematopoietic clones, such as those found in clonal hematopoiesis of indeterminate potential, may represent another hurdle to overcome [37,38], along with the fact that cfDNA could be enriched from other pathological non-tumoral tissues [39,40], hence adding another confounding factor in comorbid patients. Furthermore, cfDNA evaluation can be distorted by acute trauma, infections, stroke, exercise, and transplantation [41,42,43,44,45], making it necessary to develop better techniques for identifying tumor-specific somatic DNA variants.

In its simplest form, ctDNA analysis can be assimilated into a particular type of genotyping technique, whose goal is to test the presence of a usually restricted number of cancer-related variants. Among the main challenges associated with the ctDNA “genotyping” we may consider: (i) low concentration of circulating DNA; (ii) the existence of fragmented ctDNA; (iii) ctDNA representing only a small fraction of the total cfDNA [46,47]. This is particularly true for those samples characterized by a low tumor burden, e.g., at MRD or early diagnosis levels. The latter point is crucial and explains why ctDNA analysis requires dedicated, and often relatively costly techniques, as opposed to conventional genotyping [48]. Schematically, we can distinguish two scenarios: in the first one, a standard NGS-based approach (e.g., whole-exome or targeted sequencing) is usually adopted using the primary tumor mass, thus identifying the mutational landscape of the tumor. This map can be subsequently used to search, a priori, at nucleotide resolution level, for all the variants that need to be monitored by ctDNA analysis: this is a typical scenario that could apply to the MRD monitoring setting. In the second one, the specific variants that are present in the sample are unknown, which may significantly increase the complexity of the analysis. This typically occurs in early detection screening protocols. In all cases, however, a classical approach based on PCR amplification of the target locus/loci followed by amplicon Sanger sequencing is simply unsuitable for ctDNA analysis, as the sensitivity of Sanger technology for variants occurring at low variant allele frequency (VAF) is too low to be of any practical use in this scenario.

ctDNA analysis typically relies on the following steps: (1) cfDNA extraction; (2) enrichment of the target regions; (3) NGS library preparation; (4) NGS sequencing. cfDNA extraction is usually performed using either spin column- or magnetic bead-based methods [49,50,51,52]. Despite the spin column-based kits being considered the first choice, when it comes to collecting cfDNA for molecular analyses, they are more expensive and have longer execution times, compared to magnetic bead-based ones [53].

The enrichment step usually involves the use of targeted gene panels, aiming to investigate disease-related highly recurrent mutated genes. Moreover, in cases when the specific target mutations are known, PCR-based enrichment can be considered as well. In both scenarios, precautions must be taken to avoid preparation artifacts such as amplification-based errors. In terms of the NGS library preparation step, since high sensitivity and specificity are major goals in virtually all ctDNA analyses, several NGS protocols have been recently developed to maximize the reliability of NGS-based ctDNA variant calling [54,55,56]; these include, for instance, CAPPSeq, iDES, and SAFE-seq techniques [57,58,59], whose detailed description goes beyond the purpose of this review. The final NGS sequencing step depends upon the amount of sequencing material, being roughly proportional to the number of profiled genes and the desired depth of the analysis [60].

In general, one of the significant advantages of NGS techniques is that they are particularly sensitive in spotting single nucleotide variants (SNVs), small insertions and deletions, and copy number variations (CNVs), while the gene fusion and exon skipping detection fall within the competence of PCR and RNA sequencing.

Currently available ctDNA assays use hybrid-capture-based or Amplicon/PCR-amplified-based methodologies and are limited to a strict selection of gene sets [61]. Ongoing studies are assessing whole genome sequencing as a cheaper, more rapid, and accessible methodology for obtaining significant information from the LBx [62].

In the context of the transcriptome evaluation, the dysregulation of microRNAs (miRNAs) plays a role in the pathogenesis of several diseases, including cancer, by taking part in post-transcriptional modification of genes related to apoptosis, stress response, mitosis, and cell differentiation [17,63,64,65,66]. MiRNAs also circulate in the bloodstream, configuring themselves as promising diagnostic biomarkers and, possibly, therapeutic targets [67,68,69].

Our knowledge about the usefulness of the circulating tumor components and the appropriate techniques to evaluate the messages they carry is still scarce and requires further studies and insightful discoveries to be embedded within the clinical practice setting.

### 2.2. CTCs

CTCs derive from tumor foci, either primary or metastatic [70,71]. They are released from the primary source in form of clusters, but they can also be found as single cells, whose correlation with overall survival and prognosis has been established [72]. CTCs could provide early information mirroring the primary tumor, such as genomic alterations [73,74], and gene and protein expression [75,76,77,78] characterizing the tumor cells. The existence of sub-clonal CTC populations has been confirmed as related to the cancer metastatic spreading, [79] leading to possible new ways of diagnosing the disease progression beforehand. CTCs’ rarity, heterogeneity, and the difficulties involved in their analysis make their clinical involvement quite challenging [80,81,82]. Overall, given the ability of CTCs to mirror the tumor of origin, they have been shown to provide useful information about the tumor itself, its metastatic spreading, as well as resistance or sensitivity to therapies [83,84,85].

CTCs detection, unfortunately, requires complex enrichment techniques, on the basis of both biological (i.e., specific antibody affinity) and physical properties (i.e., selecting them through size or deformability) [86,87]. Similarly to cfDNA, the amount of CTCs compared to the number of normal nucleated blood cells is very low: this has represented a major challenge that could affect the reliability of the results [88].

For CTCs detection and counting, FDA has currently approved one platform (CellSearch^®^ system, Menarini Silicon Biosystems, Firenze, Italy), based on EpCAM^+^ CTCs identification [89].

### 2.3. Methylation Markers

DNA methylation has been reported to support the pathogenesis and disease progression of several cancer types. For instance, it can enhance the metastatic process, acting on promoter regions related to tumor suppressor genes [90,91,92]. It could also give information about the treatment response [93]. Since the methylation process could anticipate the manifestation of the full-blown disease [94], it is reasonable to think to integrate ctDNA methylation sequencing for early diagnosis, in a disease-related targeted way [95]. Based on this idea, Liu et al. conducted a study with a wide cohort of neoplastic patients, in which this method managed to diagnose more than 50 cancer types across all stages, with great specificity [96], but with low sensitivity level, particularly in early-stage cases [97]. The high specificity levels were justified by the rarity of DNA methylation findings in healthy samples’ CTCs and cfDNA [91,92,98,99]. In parallel, some studies have reported on the use of DNA methylation as a potential novel cancer biomarker, as shown for GSTPI, PITX2, and MGMT, within the context of prostate cancer, lymph node-negative breast cancer, and glioblastomas, respectively [100].

### 2.4. Extracellular Vesicles

Cells are capable of releasing vesicles across the extracellular space, as demonstrated within the context of both physiological and pathological conditions. These are heterogeneous particles, delimited by a phospholipid bilayer, responsible for several functions such as cell-to-cell communication [101], thus supporting the cross-talk between malignant cells and the microenvironment cells [102], as it has been shown within the specific context of multiple myeloma [103]. It is possible that, even when CTCs are below the threshold of detection, extracellular vesicles (EVs), for instance, nanovesicles, could be analyzed [104], and used to support the identification of cancer biomarkers. The content of these nanovesicles, together with lipids, proteins, and metabolites, often consists of small fragments of RNA and DNA [105]. They can generate from a tumor source, thus reflecting the primary tumor cell-specific mutational status, serving as a diagnostic biomarker [106,107]. In contrast to ctDNA, shed through cellular death mechanisms, EV-DNA is actively and selectively released by subclonal tumor cells and is wrapped in a membrane that protects the nucleic acids from a faster degradation, which makes it a more accurate depiction of the tumor and its diversified environment; nevertheless, the technological limitation in isolating EVs and analyzing them singularly prevents, at the moment, this promising opportunity from being adopted within clinical practice [108].

The dysregulation of microRNAs (miRNAs) plays a role in the pathogenesis of several diseases, including cancer, by taking part in post-transcriptional modification of apoptosis-, stress response-, mitosis-, and cell differentiation-related genes [17,63,64,65,66]. miRNAs circulate in the bloodstream both freely and as exosomal miRNAs, configuring themselves as promising diagnostic biomarkers and, possibly, therapeutic targets [67,68,69,109,110]. As reported in the literature, exosomal small RNAs (such as miRNAs) represent the main content of EVs [103,111,112].

EVs are usually characterized in terms of both morphology and content; this is achieved using several techniques, such as fluorescence-based platforms, electron microscopy, immunogold labeling, flow cytometry, and mass spectrometry [113,114], with great sensitivity and specificity degrees [103,115]. In the specific case of exosome characterization, for instance, CD63 and CD81 represent key markers to be investigated at the protein level [103,116,117].

## 3. Liquid Biopsy in Lymphoproliferative Diseases

Acute lymphocytic leukemia (ALL), chronic lymphocytic leukemia (CLL), lymphomas, and plasma cells dyscrasias, such as multiple myeloma (MM) and Waldenström macroglobulinemia (WM) [118] belong to the group of lymphoproliferative disorders.

cfDNA studies in hematology go back to 1994, with a specific interest in the myeloproliferative disease, including both acute myeloid leukemia (AML) and myelodysplastic syndrome (MDS). Ten years later, peripheral blood-based LBx was thought to be a potential complementary approach for investigating bone marrow (BM) biopsies or aspirates [119,120], exposing patients to a less invasive procedure and allowing clinicians and scientists to take into account spatial and cellular tumor heterogeneity [121]. Nowadays, multiple research studies are being conducted, but we are far from the complete integration of an LBx-based approach within the setting of clinical practice (Table 2).

### 3.1. Diagnosis and Prognosis

Specific ctDNA fragments and total cfDNA quantities have been reported to correlate with the presence of the disease, as demonstrated in lymphoma, in particular Hodgkin lymphoma (HL), and pediatric ALL [122,123,124]. Hohaus et al. found that mean cfDNA levels in HL and non-Hodgkin lymphoma patients were significantly higher, as compared to healthy controls [125].

Moreover, ctDNA quantitative analysis could support clinicians in patient risk stratification: Kurtz et al. developed the continuous individualized risk index (CIRI), as a prognostic scale that helps differentiate DLBLC patients with low- and high-risk diseases [126].

ctDNA (targeting IgH gene rearrangements) is found to correlate with DLBCL prognostic indexes, such as LDH, IPI score, and PET/CT-assessed tumor burden [127,128]. A similar association was also discovered in the context of MM patients [129,130].

Circulating miRNAs have been shown to identify certain B-cell malignancies. For instance, miRNA-155 blood levels are significantly increased in DLBCL, CLL, and WM patients compared to healthy controls [131,132,133], suggesting its possible applicability in diagnosing the disease.

Many studies have confirmed that EVs represent sensitive diagnostic tools supporting the diagnosis of hematologic malignancies, such as CLL, AML, HL, WM, and MM [134,135,136,137,138,139,140]. It has been shown that EVs allow for discrimination between monoclonal gammopathy of undetermined significance (MGUS), MM, and healthy subjects [141,142,143,144]. MM patients’ CD38^+^ EVs are more abundant compared to smoldering and MGUS patients, and, interestingly, correlate positively with the clinical International Staging System (ISS) [142], similarly to CD203 a^+^, CD73^+^, CD157^+^, and CD39^+^ EVs, which show different correlation types with ISS and the bone marrow plasma cell infiltration [145]. Moreover, the number of CD138^+^ EVs was found to be higher in MM patients compared to healthy controls and associated with therapy response, disease stage [144,146], and the number of bone lytic lesions [147]. In a recent study, other surface EV signatures were investigated: for instance, p-gp^−^, CD34^−^, and phosphatidylserine-positivity, not only correlated with disease progression and treatment resistance, but often expressed on CD138^−^ EVs, in accordance with the fact that the immature phenotype of CD138 ^dim^ MM cells represents a negative prognostic index and predicts the occurrence of drug resistance mechanisms [148]. 

In HL and NHL, surface EV markers are related to the lymphoma subtype and the disease stage, particularly CD30 and CD19 [139,142,149]. CD30-positive EVs were shown to correlate with unfavorable outcomes [150], making it useful to stratify lymphoma patients’ risk. Similar findings have also been observed for serum-derived CD19^−^, CD20^−^, CD52^−^, and CD37-positive EVs in CLL patients, correlating with disease stage and prognosis [135,142,151]. In particular, CD19 and CD37 are significantly elevated in the advanced stages of the disease; and related to the tumor burden [135]. In addition, CD52 could represent a marker devoted to disease progression and disease relapse [151].

The role of miRNAs in supporting the pathogenesis of hematologic malignancies has been clearly shown. For instance, miR15 a and miR16 regulate tumor proliferation in MM [140]; miR-155 was found to be deregulated in serum EVs from AML and WM patients [152]; while miR-150, miR-155, and miR-29 a-c were upregulated in CLL patients [110], suggesting their role in supporting disease biology.

Some types of EV-miRNA represent disease progression biomarkers and their levels rose in relapsed classical HL patients after treatment [139].

In an MM-focused study, the expression level of several serum exosomal miRNAs (i.e., let-7 c-5 p, miR-185-5 p) were significantly different among patients with smoldering MM, active MM subjects, and healthy individuals, thus suggesting a biological role for these EV molecules in supporting disease progression [153,154,155]. Moreover, EV-let-7 b, miR-16, and miR-18 a were revealed to predict negative PFS and OS in MM patients [156,157], thus providing evidence for their role in supporting prognosis and risk stratification.

EV count could be considered a significant diagnostic and prognostic parameter. Indeed, CLL patients present a higher number of EVs in their peripheral blood compared to healthy individuals, in direct proportion to the Rai clinical stage of the disease and, possibly, with the potential sensitivity to ibrutinib treatment. Similar results were found in WM patients, with a correlation with International Prognostic Scoring System (IPSS) [135,142]. In addition to exosomal miRNA content, exosomal messenger RNA (mRNA) could provide powerful insights about prognosis in patients suffering from onco-hematologic diseases. A trial on B cell lymphoma reported a correlation between a poor prognosis, the presence of exosomal AKT mRNA in refractory to rituximab patients, and exosomal BCL2/6 mRNA discovered at diagnosis [158].

Other tumor-derived components, such as proteins, could provide useful information. Serum proteins, such as CERU, CLUS, and THRB, were found to be upregulated in pediatric ALL patients [159], as well as E2 A in B-cell CLL patients [160].

### 3.2. Follow-Up: MRD and Relapse Settings

LBx could become a convenient and repeatable way to assess MRD, thus helping clinicians with the management of hematologic cancer patients [161,162,163,164]. Despite the small quantity of exosomal mRNA, the persistent expression of BCL2, BCL6, and MYC in EV transcripts of relapsed NHL patients, implied that mRNA could be a promising molecular detector of MRD and a sign of a poor prognosis in hematologic malignancies [156,165]. Studies have shown a correlation existing between treatment response, prognosis, MRD evaluation, and genotyping-derived profiling assays in lymphoid malignancies [166,167,168,169,170,171,172,173,174].

Multiparameter flow cytometry and molecular analysis represent the gold standards in monitoring MRD [141,175]. In light of the strong presence of EVs in the peripheral blood, tumor-derived EVs could be suitable for MRD monitoring compared to cell-based analysis [141]. In a case-control trial, van Eijndhoven et al. compared plasma EVs in a cohort of HL patients, before, during, and after treatment, discovering that several EV-miRNA levels were higher both at the time of diagnosis and at relapse, being lower in cases of complete response, as confirmed with FDG-PET [139]. In addition, HL patients showed high quantities of EV-ADAM10, a potential biomarker of disease and immunotherapy resistance, given that its activity hinders anti-tumoral immune response and hampers antibody-drug conjugates’ (ADCs) effects [176].

Among biological processes that drive cancer pathogenesis, neovascularization certainly plays a crucial role both in solid tumors and hematologic malignancies, representing an indirect way to track disease progression. In CLL, for instance, endothelial progenitor cells (EPCs) and circulating endothelial cells (CECs) increase their levels in concomitance with neovascularization events [177,178] and are predictive of aggressive disease [177]. Moreover, in patients suffering from lymphoma and MM, EPCs have been shown to correlate with angiogenesis, disease progression, and treatment outcome [179,180,181,182]. 

Decreases in ctDNA in lymphoma patients have been shown to reflect a response to therapy, and to predict an improved event-free survival at 24 months in DLBCL patients undergoing first-line therapy regimens or salvage therapy [156]. Similar findings have also been confirmed in cohorts of HL and MCL patients [183,184]. Scherer et al. tested the ctDNA levels in patients with relapsed/refractory DLBCL, unveiling their absence in those cases who achieved complete remission status [185]. Importantly, a study on 126 DLBCL patients confirmed a positive predictive value of 88,2% and a negative predictive value of 97,8% for disease relapse, underlining the role of ctDNA in predicting disease recurrences [128]. Camus et al. demonstrated that ctDNA mutational status could better detect MRD in classical HL patients who received chemotherapy compared to PET-TC [186].

The investigation of specific mutations in ctDNA may lead to the early detection of relapsed patients: for instance, the detection of NRAS, KRAS, and BRAF mutational status could identify relapsed/refractory disease and predict the outcome in terms of PFS, in patients with MM [187,188,189].

V(D)J rearrangements in ctDNA have been proven to be useful since their levels correlate with the disease’s clinical activity [190]. Biancon et al. identified clonal IgH gene rearrangements and established a proportion between treatment response, lower levels of cfDNA, and a higher PFS in patients suffering from MM [129]. The detection of CNVs also led to interesting results: in a study on MM patients, treatment responders and those who showed the stable disease had a cfDNA tumor fraction of <0.05 compared to 62% of relapsed patients [191].

In the MM context, many studies have been carried out investigating relapse and disease progression: CTCs have been shown to correlate with the spread of extramedullary lesions [192,193,194,195], and upregulated serum levels of miRNA-20 a and miRNA-148 a were able to identify patients prone to disease relapse [196].

Lastly, ctDNA methylation analyses have been performed in patients suffering from lymphoproliferative diseases [197]. Shi et al. evaluated CpG island DNA methylation profiling in a cohort of NHL patients. The methylation of the DLC-1 gene was detected in patients’ blood; moreover, methylation levels significantly declined after response to chemotherapy [198], thus suggesting the possible advantages in monitoring the disease and relapse events.

### 3.3. Drug Sensitivity and Resistance

Early prediction of drug resistance may be possible thanks to EV miRNAs, whose potential applications have been investigated mainly within the MM and lymphoma setting [138,199]. Zhang et al. demonstrated the possibility of creating a predictive biomarkers panel for bortezomib resistance in MM patients, based on the downregulation of exosomal miRNAs [200]. In B-cell lymphoma, CD20^+^ EVs may provide information about rituximab efficacy: it was demonstrated that CD20^+^ exosomes were able to bind to rituximab molecules, decreasing the active blood fraction of the circulating drug [158]. DLBCL patients presented with EV-miRNA-99 a-5 p up-regulation, and miRNA-125 b-5 p expression was shown to be correlated with shorter PFS and drug resistance [201].

Some mutations detectable in cfDNA could also give information about drug sensitivity/resistance, as in the case of MYD88 variants that were reported to predict response to ibrutinib, in DLBCL patients [202,203]. Importantly a high concordance rate between peripheral blood and BM cancer cells, in terms of both MYD88 and CXCR4 mutational status, has been clearly demonstrated in WM patients, in terms of both disease status and drug resistance [204,205,206,207,208,209,210,211].

Interestingly, plasma liposomes could also play a prognostic role and be used as biomarkers of drug resistance. For example, in MCL patients, lipid metabolism affects the apoptosis of cancer cells, and the lipid count reflects the expression of CD36, indicating lower bortezomib sensitivity [212].

## 4. Conclusions

Novel individualized approaches for the management of blood cancer patients are certainly needed. Despite the growing evidence of the usefulness of LBx, its indirect indicators are far from being completely explored, and their role is not completely known. Technological limitations are also responsible for the controversial reliability of current assessment methods. In this perspective, the emergence of novel biomarkers and improved “magnifying glasses” are likely to facilitate LBx slowly sliding into clinal practice. As for the onco-hematology setting, preliminary results are encouraging about possible future applications in preventive medicine, personalized treatments, and management of patients, particularly as a vital support for MRD detection.

Current research is focusing on increasing accuracy levels, the discovery of new biomarkers, detection and isolation techniques, and more specific analysis methods. The greater the means, the wider the possibilities of implementing LBx in practical protocols.

Although experimental data results are promising, the question of whether LBx integration in clinical practice is feasible remains to be fully answered. Lastly, before broadly embedding LBx as a routine test in medical oncology, it is still necessary to demonstrate its potential contribution to patients’ outcomes.

## Figures and Tables

**Figure 1 cancers-14-05378-f001:**
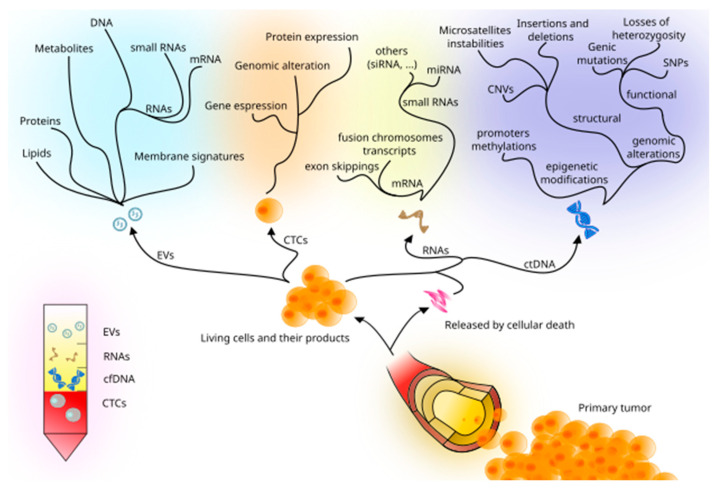
Principal tumor-derived elements detectable in peripheral blood and the assessable information they may carry.

**Table 1 cancers-14-05378-t001:** Pros and cons of LBx compared to solid tissue biopsy.

Pros	Cons
Constant tissue production	Costs
Fast process	Not reliable yet
No unfit patients	Several confounding factors
Fewer complications	Low availability of specialized machine that would improve the accuracy
Simple procedure	Not all components are detectable in early and MRD settings
Easy repeatability	-
Subclonal genetic makeup	-
Metastasis data	-
Possibility of observing evolution over time	-
More accurate with regard to prognosis and treatment response	-
Early diagnosis, even without knowing the primary tumor location	-

**Table 2 cancers-14-05378-t002:** Ongoing studies on LBx in hematologic malignancies, provided by the ClinicalTrials.gov search engine, current as of the date of this article.

Target	Technology Used	ClinicalTrials.gov IDs	Study Title	Study Endpoints
ctDNA	Genetic tests NOS	NCT03023202	UWCCC Precision Medicine Molecular Tumor Board Registry	Frequency of acceptance of molecular tumor board recommendations. Benefits of PMMTB-recommended treatment. Correlation of mutations with protein overexpression, circulating tumor DNA, and spheroid culture investigations.
ctDNA, RNA	NGS, RNA sequencing	NCT01775072	Genomic Profiling in Cancer Patients	Nature and the frequency of “actionable” oncogenic mutations.
miRNA	PCR	NCT02791217	Identification of Hematological Malignancies and Therapy Predication Using microRNAs as a Diagnostic Tool	Molecular characteristics (GEP, miRNA); EFS; OS.
ctDNA, RNA	Genetic tests NOS	NCT01792882	Prospective Collection of Surplus Surgical Tumor Tissues and Pre-surgical Blood Samples	Tumor genetic sequence variation. Transcription profile. Epigenetic modification.
ctDNA	Genetic tests NOS	NCT01137643	Tissue, Blood, and Body Fluid Sample Collection from Patients With Hematologic Cancer	Development of a centralized, quality-controlled, quality-assured facility for the procurement, processing, storage, and distribution of normal and malignant tissue specimens and corresponding blood specimens.
ctDNA, RNA	NGS, RNA sequencing	NCT02213822	Molecular Testing of Cancer by Integrated Genomic, Transcriptomic, and Proteomic Analysis	Frequency of “actionable” oncogenic mutations; prevalence of genomic, transcriptomic, and proteomic abnormalities.
Omics	Genetic tests NOS	NCT04298892	Integrated Multiomics and Multilevel Characterization of Haematological Disorders and Malignancies	Hematologic diseases characterization. Response/resistance to ex vivo drug treatments. Biomarkers of drug-related toxicity. Association between biological and molecular features with patient’s clinical features. MRD. Recurrence/MRD patterns after treatments. Prognostic and early diagnostic biomarkers. Identification of circulating and tissue molecular markers. Technological advancement.
Epigenomics	Genetic tests NOS	NCT04264767	Characterization of Methylation Patterns in Cancer and Non-Cancer cfDNA	Characterization of methylation patterns that will discriminate cancer and normal samples and the origin of cancer.
ctDNA	Genetic tests NOS	NCT01772771	Molecular Testing for the MD Anderson Cancer Center Personalized Cancer Therapy Program	Frequency and distribution of mutations and co-mutations between different tumor types and levels of clinical-pathological factors.
ctDNA, epigenomics	Genetic tests NOS	NCT03727009	Blood Sample Collection to Evaluate Biomarkers in Subjects with Untreated Hematologic Malignancies	Blood-based biomarkers associated with genetic and epigenetic alterations.
CTCs	Magnetic nanoparticles coated with antibodies	NCT04290923	Determination of Blood Tumor Cells	CTC counting
ctDNA	NGS	NCT02534649	Bergonie Institut Profiling: Fighting Cancer by Matching Molecular Alterations and Drugs in Early Phase Trials	Efficacy of LBx in terms of frequency of genomic alteration, molecular profiling, failure rate of molecular screenings, and safety of the procedures.

NOS: not otherwise specified. PMMTB: Precision Medicine Molecular Tumor Board. EFS: event-free survival. OS: overall survival. GEP: gene expression profiling. MRD: minimal residual disease. CTC: circulating tumor cells.

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
