# Peer review of "Liquid Biopsy in Cancer: Focus on Lymphoproliferative Disorders"

_cancers, 2022, doi:10.3390/cancers14215378_

Round 1

Reviewer 1 Report

Recommendation: Publish after major revisions noted.  

Comments:  

This manuscript highlights liquid biopsy in cancer. The authors need to address the following comments and revise the manuscript accordingly. 

1. This manuscript is in need of substantial editing, English language and style improvement.

2. Introduction: Page 2, line 49: Consider to combine those multiple paragraphs and avoid plagiarism. Describe “liquid biopsy” briefly and the potential of liquid biopsy for early detection of cancer. Consider to include the following references: 

a) Roy, D.; Pascher, A.; Juratli, M.A.; Sporn, J.C. The Potential of Aptamer-Mediated Liquid Biopsy for Early Detection of Cancer. Int. J. Mol. Sci. 2021, 22, 5601. https://doi.org/10.3390/ijms22115601 

b) Valihrach L, Androvic P, Kubista M. Circulating miRNA analysis for cancer diagnostics and therapy. Mol Aspects Med. 2020 Apr;72:100825. doi: 10.1016/j.mam.2019.10.002. Epub 2019 Oct 18. PMID: 31635843. 

c) Roy D, Lucci A, Ignatiadis M, Jeffrey SS. Cell-free circulating tumor DNA profiling in cancer management. Trends Mol Med. 2021 Jul 23:S1471-4914(21)00182-9. doi: 10.1016/j.molmed.2021.07.001. 

3. Page 10, line 430: Table 2, consider adding columns for the technology utilized and the findings of those trials.

4. CfDNA is one of the key components in liquid biopsy based detection. Highlight ctDNA and please consider to include DNA methylation based assays for early detection of cancer. Consider to add the following references. 

 Duffy MJ, Napieralski R, Martens JWM et al. Methylated genes as new cancer biomarkers. Eur J Cancer 2009; 45(3): 335–346.

5. Please consider include a section on the limitations of liquid biopsy and the challenges for clinical applications. 

Author Response

REVIEWER #1
Q1. This manuscript highlights liquid biopsy in cancer. The authors need to address the following
comments and revise the manuscript accordingly. This manuscript is in need of substantial editing,
English language and style improvement.

A1. We have revised the text, in terms of both English language and style. Revisions are being tracked in
red, throughout the manuscript.

Q2. Introduction: Page 2, line 49: Consider to combine those multiple paragraphs and avoid plagiarism.
Describe “liquid biopsy” briefly and the potential of liquid biopsy for early detection of cancer.
Consider to include the following references:

a) Roy, D.; Pascher, A.; Juratli, M.A.; Sporn, J.C. The Potential of Aptamer-Mediated Liquid
Biopsy for Early Detection of Cancer. Int. J. Mol. Sci. 2021, 22, 5601.
https://doi.org/10.3390/ijms22115601;

b) Valihrach L, Androvic P, Kubista M. Circulating miRNA analysis for cancer diagnostics and
therapy. Mol Aspects Med. 2020 Apr;72:100825. doi: 10.1016/j.mam.2019.10.002. Epub
2019 Oct 18. PMID: 31635843;

c) Roy D, Lucci A, Ignatiadis M, Jeffrey SS. Cell-free circulating tumor DNA profiling in cancer
management. Trends Mol Med. 2021 Jul 23:S1471-4914(21)00182-9. doi:
10.1016/j.molmed.2021.07.001.

A2. Manuscript has been revised according to Reviewer’s comment. Specifically, text has bene modified
in order to:

- combine some of the paragraphs provided within the introduction section (page 2);

- provide a brief description of liquid biopsy (lines 49-56) and its potential in terms of cancer
early detection (lines 69-71; 214-2019);

- Include the references mentioned above (#18, #19, #21).

Q3. Page 10, line 430: Table 2, consider adding columns for the technology utilized and the findings of
those trials.

A3. Table 2 has been revised according to Reviewer’s comment: ad additional columns was added in
order to provided details on the technology utilized for each given trial. We agree on the importance
of including also the findings related to each trial: in this regard, we would need to point out that
most of the trials are being conducted and, therefore, it would not be possible to refer to trial-related
findings. We would kindly ask the Reviewer to consider the option not to include the column named
“findings”.

Q4. CfDNA is one of the key components in liquid biopsy based detection. Highlight ctDNA and please
consider to include DNA methylation based assays for early detection of cancer. Consider to add
the following references: Duffy MJ, Napieralski R, Martens JWM et al. Methylated genes as new
cancer biomarkers. Eur J Cancer 2009; 45(3): 335346.

A4. Manuscript has been revised according to Reviewer’s comment, and the reference mentioned above
has been included (lines 210-224).

Q5. Please consider include a section on the limitations of liquid biopsy and the challenges for clinical
applications.

A5. Manuscript has been revised according to Reviewer’s comment and included the reference to issues
related to liquid biopsy (lines 100-104; 139-141; 2015-207).

REVIEWER #2
The authors provided a well-documented and brilliantly organized overview of the potential impact of
liquid biopsy options for hematological tumors management. In addition to this, the review could
represent a really interesting point of view in a field so dynamic and rich in potential future applications.

Q1. If the article is well written, the introduction section could be improved by including more detailed
argumentations regarding EVs: new technologies for the association of a specific marker with an
EVs subtype and the EVs subtype to a particular function and/or group of functions it's becoming
vital (PMID: 35141731 and others). I hope that my comments could be useful and I look forward
to reading the revised version of the paper.

A1. Manuscript has been revised according to Reviewer’s comment. We agree on the importance of
extracellular vescicles (EVs) and their potential to be used as novel cancer biomarkers, as detailed
within paragraph #2.4. However, the introduction section was not structured to refer to this specific
topic: introduction has been revised accordingly (lines 86-93)

Reviewer 2 Report

The authors provided a well-documented and brilliantly organized overview of the potential impact of liquid biopsy options for hematological tumors management.  In addition to this, the review could represent a really interesting point of view in a field so dynamic and rich in potential future applications.  If the article is well written, the introduction section could be improved by including more detailed argumentations regarding EVs: new technologies for the association of a specific marker with an EVs subtype and the EVs subtype to a particular function and/or group of functions it's becoming vital (PMID: 35141731 and others).  I hope that my comments could be useful and I look forward to reading the revised version of the paper.

Author Response

POINT-BY-POINT ANSWERS TO REVIEWERS’ COMMENTS
REVIEWER #1

Q1. This manuscript highlights liquid biopsy in cancer. The authors need to address the following
comments and revise the manuscript accordingly. This manuscript is in need of substantial editing,
English language and style improvement.

A1. We have revised the text, in terms of both English language and style. Revisions are being tracked in
red, throughout the manuscript.

Q2. Introduction: Page 2, line 49: Consider to combine those multiple paragraphs and avoid plagiarism.
Describe “liquid biopsy” briefly and the potential of liquid biopsy for early detection of cancer.
Consider to include the following references:

a) Roy, D.; Pascher, A.; Juratli, M.A.; Sporn, J.C. The Potential of Aptamer-Mediated Liquid
Biopsy for Early Detection of Cancer. Int. J. Mol. Sci. 2021, 22, 5601.
https://doi.org/10.3390/ijms22115601;

b) Valihrach L, Androvic P, Kubista M. Circulating miRNA analysis for cancer diagnostics and
therapy. Mol Aspects Med. 2020 Apr;72:100825. doi: 10.1016/j.mam.2019.10.002. Epub
2019 Oct 18. PMID: 31635843;

c) Roy D, Lucci A, Ignatiadis M, Jeffrey SS. Cell-free circulating tumor DNA profiling in cancer
management. Trends Mol Med. 2021 Jul 23:S1471-4914(21)00182-9. doi:
10.1016/j.molmed.2021.07.001.

A2. Manuscript has been revised according to Reviewer’s comment. Specifically, text has bene modified
in order to:

- combine some of the paragraphs provided within the introduction section (page 2);

- provide a brief description of liquid biopsy (lines 49-56) and its potential in terms of cancer
early detection (lines 69-71; 214-2019);

- Include the references mentioned above (#18, #19, #21).

Q3. Page 10, line 430: Table 2, consider adding columns for the technology utilized and the findings of
those trials.

A3. Table 2 has been revised according to Reviewer’s comment: ad additional columns was added in
order to provided details on the technology utilized for each given trial. We agree on the importance
of including also the findings related to each trial: in this regard, we would need to point out that
most of the trials are being conducted and, therefore, it would not be possible to refer to trial-related
findings. We would kindly ask the Reviewer to consider the option not to include the column named
“findings”.

Q4. CfDNA is one of the key components in liquid biopsy based detection. Highlight ctDNA and please
consider to include DNA methylation based assays for early detection of cancer. Consider to add
the following references: Duffy MJ, Napieralski R, Martens JWM et al. Methylated genes as new
cancer biomarkers. Eur J Cancer 2009; 45(3): 335346.

A4. Manuscript has been revised according to Reviewer’s comment, and the reference mentioned above
has been included (lines 210-224).

Q5. Please consider include a section on the limitations of liquid biopsy and the challenges for clinical
applications.

A5. Manuscript has been revised according to Reviewer’s comment and included the reference to issues
related to liquid biopsy (lines 100-104; 139-141; 2015-207).

REVIEWER #2
The authors provided a well-documented and brilliantly organized overview of the potential impact of
liquid biopsy options for hematological tumors management. In addition to this, the review could
represent a really interesting point of view in a field so dynamic and rich in potential future applications.

Q1. If the article is well written, the introduction section could be improved by including more detailed
argumentations regarding EVs: new technologies for the association of a specific marker with an
EVs subtype and the EVs subtype to a particular function and/or group of functions it's becoming
vital (PMID: 35141731 and others). I hope that my comments could be useful and I look forward
to reading the revised version of the paper.

A1. Manuscript has been revised according to Reviewer’s comment. We agree on the importance of
extracellular vescicles (EVs) and their potential to be used as novel cancer biomarkers, as detailed
within paragraph #2.4. However, the introduction section was not structured to refer to this specific
topic: introduction has been revised accordingly (lines 86-93).

Round 2

Reviewer 1 Report

The authors have addressed the comments quite thoroughly and this version of the manuscript is improved. Please publish after minor revisions.

1.  Page 10, line 430: Table 2, consider adding columns for the technology utilized and the findings of those trials. 

Please include the primary and secondary endpoints of any studies that are still ongoing. 

Author Response

Please, refer to the attached file (Answer to Reviewer_2nd round)
